# LncRNA MALAT1 Regulates Hyperglycemia Induced EMT in Keratinocyte via miR-205

**DOI:** 10.3390/ncrna9010014

**Published:** 2023-02-11

**Authors:** Liping Zhang, George Chu-Chih Hung, Songmei Meng, Robin Evans, Junwang Xu

**Affiliations:** 1Department of Physiology, College of Medicine, University of Tennessee Health Science Center, Memphis, TN 38163, USA; 2Division of Plastic Surgery, Department of Surgery, College of Medicine, University of Tennessee Health Science Center, Memphis, TN 38163, USA

**Keywords:** diabetic wounds, long non-coding RNA, MALAT1, epithelial mesenchymal transition

## Abstract

Epithelial-to-mesenchymal transition (EMT) is critical to cutaneous wound healing. When skin is injured, EMT activates and mobilizes keratinocytes toward the wound bed, therefore enabling re-epithelialization. This process becomes dysregulated in patients with diabetes mellitus (DM). Long non-coding RNAs (lncRNAs) regulate many biological processes. LncRNA-metastasis-associated lung adenocarcinoma transcript 1 (MALAT1) influences numerous cellular processes, including EMT. The objective of the current study is to explore the role of MALAT1 in hyperglycemia (HG)-induced EMT. The expression of MALAT1 was found to be significantly upregulated, while the expression of miR-205 was downregulated in diabetic wounds and high-glucose-treated HaCaT cells. The initiation of EMT in HaCaT cells from hyperglycemia was confirmed by a morphological change, the increased expression of CDH2, KRT10, and ACTA2, and the downregulation of CDH1. The knockdown of MALAT1 was achieved by transfecting a small interfering RNA (SiRNA). MALAT1 and miR-205 were found to modulate HG-induced EMT. MALAT1 silencing or miR-205 overexpression appears to attenuate hyperglycemia-induced EMT. Mechanistically, MALAT1 affects HG-induced EMT through binding to miR-205 and therefore inducing ZEB1, a critical transcription factor for EMT. In summary, lncRNA MALAT1 is involved in the hyperglycemia-induced EMT of human HaCaT cells. This provides a new perspective on the pathogenesis of diabetic wounds.

## 1. Introduction

Diabetes affects more than 11 percent of Americans, making it one of the most significant health concerns of the 21st century (2022 National Diabetes Statistics Report). Impaired wound healing is a major complication among diabetic patients. Despite recent progress, the pathogenesis of diabetic wounds remains unclear. Currently, the prognosis of a diabetic wound is very poor [1]. Diabetes is a state of systemic hyperglycemia that may influence wound healing through the alteration of genetic expression. Through improved understanding of the underlying molecular mechanisms, more effective pharmacological therapies may be developed. 

Optimal cutaneous wound healing is an overlapping multi-step process that includes hemostasis, inflammation, re-epithelialization, granulation tissue formation, neovascularization, and remodeling [2]. Within each step, there is a complex interplay between cell types, cytokines, and gene regulation. Keratinocytes are one of the most abundant epithelial cells. Upon injury, they migrate from the wound edge into the wound to re-epithelialize damaged tissues [3]. This response is disrupted in diabetic patients. One of the hallmarks of a non-healing diabetic wound is impaired re-epithelialization [4]. Under hyperglycemia, keratinocytes show a restrained migration capability, leading to poor re-epithelialization and wound closure [5,6]. Understanding the factors contributing to EMT in the context of hyperglycemia will enable novel therapeutic targets for non-healing diabetic wounds.

Long non-coding RNAs (lncRNA) are a class of non-protein coding RNAs longer than 200 nucleotides in length. They are recognized as important regulators in the development and pathophysiology of disease [7,8]. Evidence regarding the role of lncRNA in diabetic wounds is now emerging [9,10]. MALAT1 is an lncRNA that was first discovered as a prognostic factor for lung cancer metastasis [11]. Recent studies have shown that MALAT1 plays an essential role in regulating EMT in various cancers via a variety of mechanisms [12,13,14,15,16,17]. In addition to carcinogenic cells, MALAT1 is also involved in EMT in non-carcinogenic cells. In endothelial progenitor cells (EPCs), MALAT1 promotes TGF-β1-induced EMT by binding to miR-145 [18]. In retinal pigment epithelium (RPE), MALAT1 activates Smad2/3 signaling and promotes TGF-β1-induced EMT [19]. MALAT1 also regulates ZEB2 by binding miR-145 and promoting EMT in hyperglycemia stimulated HK-2 cells [20]. These studies indicate that MALAT1 has the potential to promote EMT through various pathways. Our recent study demonstrated that MALAT1 is involved in TGF-β1-induced EMT in human keratinocyte (HaCaT) cells via upregulating ZEB1 [21]. Meanwhile, other studies have shown that MALAT1 can be upregulated under hyperglycemia through multiple pathways [22,23,24]. However, whether MALAT1 is required for EMT under hyperglycemia in keratinocytes has yet to be elucidated.

Mounting evidence has shown that microRNAs (miRNAs) also play a crucial role in EMT [25]. The miRNA miR-205 has been found to possess an inhibitory effect on EMT. Previous studies have shown that miR-205 suppresses EMT through an inverse relationship with E-cadherin transcriptional repressors ZEB1 and ZEB2 [26,27]. 

The aim of the present study is to investigate the roles of MALAT1 and miR-205 in EMT under hyperglycemic conditions in keratinocytes. Diabetic wounds and high-glucose-treated cells were included in our experiments to examine the factors of EMT during hyperglycemia. We hypothesize that the lncRNA MALAT1 binds to miR-205 to elevate the expression of the target gene ZEB1, thereby inducing EMT under hyperglycemia conditions.

## 2. Materials and Methods

### 2.1. Cells Culture Conditions and Treatments

HaCaT cells, obtained from ATCC, were cultured in full medium comprising Dulbecco’s modified eagle high-glucose (DMEM, Sigma-Aldrich, St. Louis, MO, USA), supplemented with 10% fetal bovine serum (FBS; Gibco, MA, USA) and maintained at 37 °C in a humidified atmosphere containing 5% CO_2_. Cells were serum-starved overnight in DMEM with 1% FBS, 100 μg/mL streptomycin, and 100 U/mL penicillin before being treated with 5 mM D-glucose (LG) DMEM and 25 mM D-Glucose (HG) for 24 h, as described [28]. SiRNA (si-MALAT1, or si-Con) transfection of HaCaT cells was performed using the Lipofectamine 2000 reagent (Invitrogen, Life Technologies) according to the manufacturer’s instructions. For overexpression, HaCaT were transfected with miR-205 mimic or mimic control. Transfection reagents, siRNA, mimic, and control miRNAs were purchased from Invitrogen. Twenty-four hours following transfection, cells were processed for gene expression analysis. 

### 2.2. Real Time Quantitative PCR

Total RNA was extracted with TRIzol reagent (Invitrogen, Carlsbad, CA, USA) according to the manufacturer’s established protocol. For gene expression analysis, RNA was converted into cDNA using the SuperScript First-Strand Synthesis System (Invitrogen, Life Technologies). CDH1, CDH2, KRT10, and ACTA2 were amplified using the TaqMan gene expression assay (Applied Biosystems). Internal normalization was achieved using the GAPDH housekeeping gene. Samples (n = 5 per group) were amplified in triplicates and results were averaged for each individual sample. The ΔΔCT method was used to calculate relative gene expression. Results are reported as mean ± SD.

### 2.3. Immunocytochemistry 

Cells were fixed with 4% paraformaldehyde for 15 min, and then permeabilized with 0.1% Triton X-100 in PBS. After washing with PBS, cells were preincubated in blocking solution (1% bovine serum albumin, BSA) in PBST for 1 h at room temperature, and then incubated with primary mouse antibody against K10 (ab76318, Abcam, Waltham, MA, USA) overnight. After rinsing with PBS, the cells were incubated with a second anti-mouse antibody labeled with Alexa 488 (Abcam, Waltham, MA, USA) for 1 h. Nuclei were stained with 4’,6-Diamidino-2-Phenylindole Dihydrochloride (DAPI). Fluorescence images were captured using a Revolve microscope and analyzed. 

### 2.4. Animal Studies

All experimental protocols were approved by the Institutional Animal Care and Use Committee at University of Tennessee Health Science Center and followed the guidelines described in the NIH Guide for the Care and Use of Laboratory Animals. Twelve-week old matched, female, genetically diabetic C57BKS.Cg-m/Leprdb/J (Db) mice and heterozygous, non-diabetic (non-Db), female controls were obtained from the Jackson Laboratory (Bar Harbor, ME) and were used in this experiment. All wounding procedures were performed under inhaled Isoflurane anesthesia. The posterior neck and back were shaved and depilated prior to wounding. The area was cleaned with an alcohol swab and a single dorsal full-thickness wound was made with an 8 mm punch biopsy (Miltex, Inc., York, PA, USA) Wounds were then dressed with a Tegaderm (3M), which was subsequently removed on post-operative day 2. A full-thickness skin sample, centered on the wound, was harvested at 3 and 7 days after surgery (n = 5 per timepoint).

### 2.5. Statistical Analysis

Results are expressed as mean ± SD for n = 3 to 5 for the number of independent experiments. Statistically significant differences in gene expression between two groups was assessed by Student’s *t*-test. *p* < 0.05 was considered to be statistically significant. Pearson correlation was used for the correlation analyses. 

## 3. Results

### 3.1. Induction of MALAT1 Expression and Reduction of miR-205 Expression under Diabetic Condition 

We compared the expression of MALAT1 and miR-205 in non-diabetic and diabetic mice wounds at day 3 after the injury. Results from a real-time qPCR (RT-qPCR) showed that the expression of MALAT1 was significantly upregulated, while miR-205 was significantly downregulated in diabetic wounds compared to non-diabetic wounds at day 3 after the injury (Figure 1A,B). A correlation analysis also indicated that MALAT1 and miR-205 were reversely correlated to each other with r = 0.94 in the non-diabetic day 3 wounds (*p* = 0.0045) and r = 0.95 in the diabetic day 3 wounds (*p* = 0.0038). 

As well, we measured the expression level of the MALAT1 and miR-205 genes in human non-diabetic (ND) and diabetic (DB) skin. The specimens were collected within eight hours of death through the National Disease Research Interchange with IRB approval. Samples were obtained from patients who were 45 to 65 years of age who had no known malignancy or history of radiation or chemotherapy [29]. RT-qPCR showed that the expression of MALAT1 was significantly upregulated in human diabetic skin, while the expression of miR-205 was significantly downregulated (Figure 1C,D). A correlation analysis indicated that MALAT1 and miR-205 were reversely correlated to each other with r = 0.95 in human non-diabetic skin and r = 0.96 in human diabetic skin, which is similar to the mice wounds.

### 3.2. Hyperglycemia Induces EMT in HaCaT Cells

We then asked whether MALAT1 and miR-205 expression would be altered in a hyperglycemic environment. HaCaT cells were cultured in low (5 mM) or high (30 mM) glucose media for 4 and 24 h. The results showed that MALAT1 gene expression was significantly upregulated in HaCaT cells treated with high-glucose media, while miR-205 gene expression was significantly reduced when treated with low-glucose media (Figure 2A,B). 

Hyperglycemia (HG)-induced EMT has been well-documented in various cells [19,30,31,32]. However, the effect of hyperglycemia on EMT in HaCaT cells has not been explored. HaCaT cells were treated with either low or high glucose. After being incubated for 24 h, the HaCaT cells underwent EMT. This was confirmed by a morphological change to a spindle-shaped cells, and a decreased expression of the epithelial marker CDH1. Furthermore, there was an enhanced expression of mesenchymal markers, including CDH2, ACTA2, KRT10, and MMP9 (Figure 3A–E). This illustrates that hyperglycemia appears to induce EMT in keratinocytes. 

### 3.3. Knockdown of MALAT1 Attenuates the Hyperglycemia-Induced EMT in HaCaT Cells

After using a knockdown of MALAT1 in HaCaT cells, we then explored the role of MALAT1 in hyperglycemia-induced EMT. HaCaT cells were transfected with a MALAT1-specific siRNA (Si-MALAT1) or a negative control siRNA (Si-Con) after being starved for 16 h. After incubating the cells with high glucose for 24 h, the expression of MALAT1 was then detected by RT-qPCR. Compared with si-Con, transfection with Si-MALAT1 decreased the expression of MALAT1 by more than 60%. We measured the expression of EMT-related genes, KRT10 and ACTA2, by RT-qPCR. It was found that the knockdown of MALAT1 significantly attenuated the high-glucose upregulation of KRT10 and ACTA2 (Figure 4C,D). Furthermore, the knockdown of MALAT1 abrogates the high-glucose-induced morphological change in HaCaT cells and KRT10-positive cells. As indicated in Figure 4A,B, the KRT10-positive cells were higher in high-glucose-treated HaCaT cells compared to low-glucose-treated HaCaT cells. However, the knockdown of MALAT1 in high-glucose-treated HaCaT cells significantly reduced the KRT10-positive cells. These results indicated that MALAT1 contributes to the HG-induced EMT of HaCaT cells.

### 3.4. Overexpression of miR-205 Inhibits the Hyperglycemia-Induced EMT in HaCaT Cells

To explore the potential mechanism of MALAT1 in regulating hyperglycemia-induced EMT, we searched the lncRNA and miRNA interactions website (https://diana.ece.uth.gr/lncbasev3/ (accessed on 2 December 2022) interactions) and found that MALAT1 highly interacts with miR-205. A bioinformatic analysis was conducted to identify the binding information. The binding site of miR-205 on the MALAT1 gene is located at 3374 to 3380 and the bind was via the miR-205 seed sequence UUACUUC. The ZEB1 gene as the miR-205 target was predicted through at least three databases (TargetScan, miR-DB, and miRbase), and we confirmed there is one binding site of miR-205 on the ZEB1 gene, which is located at 818 to 825 at its 3′ UTR through miR-205 seed sequence UACUUCCU. In order to understand the role of miR-205 on EMT progression induced by hyperglycemia, HaCaT cells were transfected with miR-205 mimic to achieve miR-205 overexpression. As showed in Figure 5A, miR-205 was highly upregulated by mimic transfection. Subsequently, we measured the expression of EMT-related genes. The overexpression of miR-205 significantly induced CDH1 gene expression, the epithelial marker gene, while it significantly reduced CDH2, TAGLN, ACTA2, and KRT10 gene expression, indicating that an overexpression of miR-205 may reverse the high-glucose-induced EMT progression. (Figure 5B–F). ZEB1 and MALAT1 were both inhibited by the overexpression of miR-205 (Figure 5G,H). In addition, ZEB1 was found to decrease when MALAT1 was being silenced (Figure 5I). This indicates that MALAT1/miR-205 regulates the EMT pathway via ZEB1. 

### 3.5. miR-205 Regulates EMT-Related Transcription Factor 

To investigate the potential target of miR-205 on the EMT progress mediated though high glucose, we detected the expression of EMT-related transcription factors under miR-205 mimic transfection in HaCaT cells. As indicated in Figure 5G, the overexpression of miR-205 significantly reduced the expression of ZEB1. These data indicated that the miR-205 mimics reduced the ZEB1 expression and impeded the EMT process in vitro. As a schematic diagram of our study, Figure 6 illustrates that MALAT1 may regulate the progression of EMT in diabetic wounds through the miR-205/ZEB1 pathway.

## 4. Discussion

The concept of EMT has recently been broadened from its role in embryonic development to wide-ranging effects on wound healing, fibrosis, and cancer metastasis [33,34]. An intermediary phase between epithelial and mesenchymal states, so-called partial EMT, has been broadly reported. Partial EMT is also associated with wound healing. During re-epithelialization, keratinocytes undergo partial EMT by activating morphological modifications and therefore enabling migration into the wound site [35]. In the present study, we investigated various factors, including lncRNAs and transcription factors, in order to better understand HG-induced EMT in keratinocytes. In human HaCaT cells, we found that MALAT1 was significantly involved in HG-induced EMT. Our results showed that EMT-induced markers, ACTA2 and KRT10, were substantially inhibited when high-glucose-treated HaCaT cells were transfected with si-MALAT1. On the other hand, when the normal HaCaT cells were stimulated with high-glucose media for 24 h, we found that four EMT-induced markers were upregulated and that the E-cadherin CDH1 was downregulated. This indicates that the EMT status of the HaCaT cells can be altered by MALAT1 under high-glucose conditions. Many studies examining HG-induced EMT in renal proximal tubular epithelial cells have been performed with a focus on the TGF-β1 relevant pathways [36,37,38]. However, studies on HG-induced EMT in keratinocytes are lacking. The present study showed that, in HaCaT cells, the mRNA expression of MALAT1 increased when treated with high glucose for 4 and 24 h. Our previous work has demonstrated that in HaCaT cells, MALAT1 is involved in TGF-β1-induced EMT [21]. Taken together, these results suggest that high glucose induces EMT, possibly via regulating MALAT1 in keratinocytes. 

To further understand the potential mechanism of MALAT1 in regulating HG-induced EMT, we investigated miR-205 and its downstream target transcription factor, ZEB1. We first found that the miR-205 reduced in HG-treated HaCaT cells on the same time points when MALAT1 increased. As mentioned above, MALAT1 has the potential to bind miR-205. This could result in a significant reduction of miR-205 [39,40]. Therefore, when miR-205 was found to decrease in HG-treated HaCaT cells, we presumed this to be due to the increase in MALAT1. Our results in the present study further showed that overexpressing miR-205 downregulated the E-cadherin transcriptional repressor ZEB1 in HaCaT cells. Supporting our hypothesis, it appears EMT is affected by hyperglycemia. Furthermore, EMT-induced markers, CDH2, TAGLN, ACTA2, and KRT10, were significantly decreased in the miR-205 overexpressing group. The fact that miR-205 was reduced by the high-glucose treatment indicates that EMT can be induced via this MALAT1/miR-205 pathway under hyperglycemia condition in keratinocytes. 

Non-coding RNAs, especially miRNAs and lncRNAs, have been emerging as important molecules in regulating gene transcription, both physiologically and pathologically [41]. Ali et al. summarized the possible roles of miRNAs in diabetes and cancer, and suggested that the miR-200 family was involved in diabetes and EMT [42]. Feng et al. reviewed the studies of several lncRNAs to understand their regulatory mechanisms in diabetes and relevant complications [43]. However, there are few studies that focus on non-coding RNAs in diabetic wound healing. Our lab has been conducting research on lncRNAs and microRNAs in diabetic wound healing in the hope of finding potential therapeutic targets [21,44]. In a wounded mouse model, MALAT1 was found to be significantly higher in the diabetic group than the control. In the same in vivo model, miR-205 was found to be significantly lower in the diabetic mouse. This ties in with our in vitro gene expression results when treating the cells in a high-glucose environment. The present study provides insight into the MALAT1/miR-205 pathway in the regulation of EMT of keratinocytes in hyperglycemic conditions. This is critical for developing a clinically relevant therapeutic target. Future studies will be conducted to further elucidate the details of MALAT1 in different diabetes and EMT phases.

## Figures and Tables

**Figure 1 ncrna-09-00014-f001:**
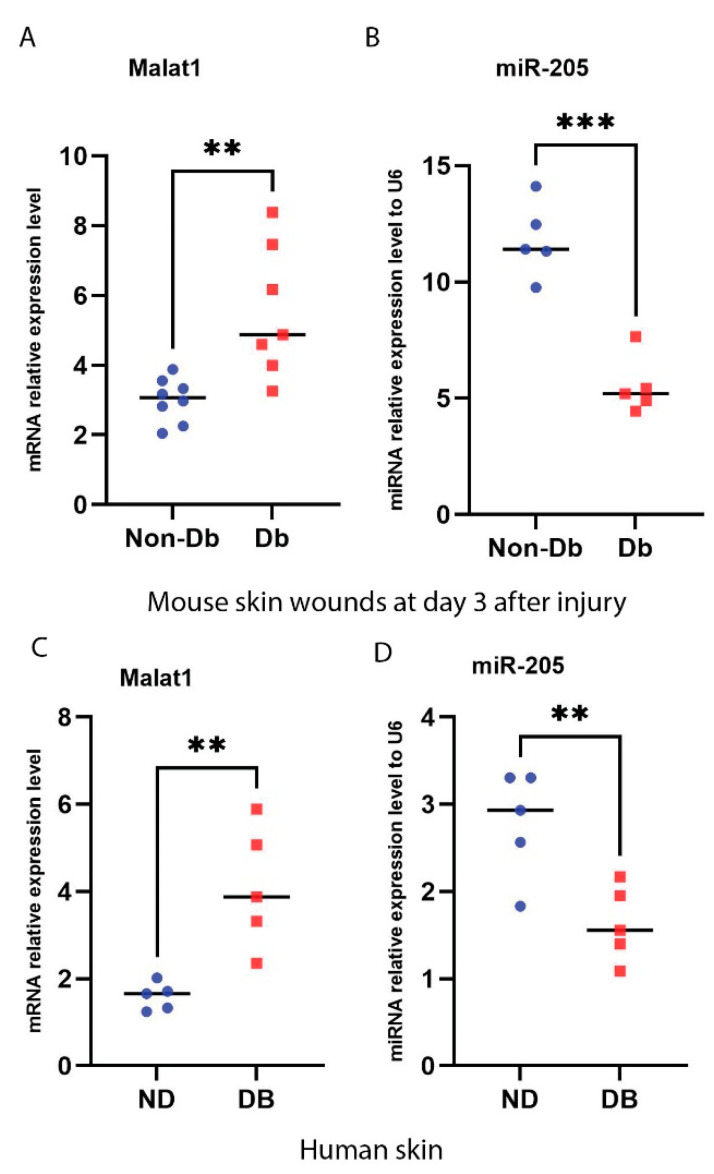
Induction of MALAT1 expression and reduction of miR-205 expression under diabetic condition. (**A**,**B**) Real-time qPCR analysis of MALAT1 and miR-205 gene expression in diabetic and non-diabetic mice wounds at day 3 (mean ± SD, n = 7 or 8 per group) after injury. (**C**,**D**) Real-time qPCR analysis of MALAT1 and miR-205 gene expression in human diabetic and non-diabetic skin (mean ± SD, n = 5 per group). ** *p* < 0.01; *** *p* < 0.001.

**Figure 2 ncrna-09-00014-f002:**
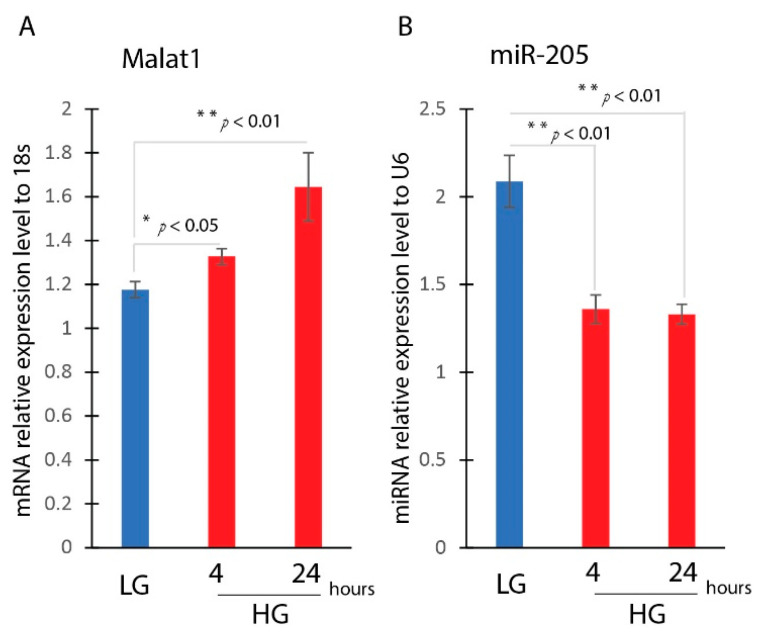
Hyperglycemia induces MALAT1 and reduces miR-205 gene expression in HaCaT cells. Real-time qPCR analysis of MALAT1 (**A**) and miR-205 (**B**) gene expression in the RAW cells treated with high glucose (25 mM D-glucose) for 4 and 24 h (mean ± SD, n = 5 per group). * *p* < 0.05, ** *p* < 0.01.

**Figure 3 ncrna-09-00014-f003:**
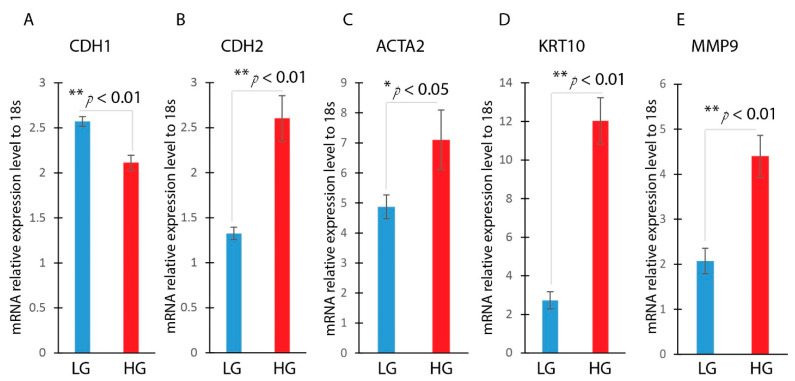
Hyperglycemia-induced EMT in HaCaT cells. HaCaT cells were treated with low (5 mM D-glucose) or high glucose (25 mM D-glucose) after overnight serum starvation. RNAs were isolated and the expressions of CDH1 (**A**), CDH2 (**B**), ACTA2 (**C**), KRT10 (**D**), and MMP9 (**E**) were determined by real-time qPCR. n = 3; mean ± SD; * *p* < 0.05, ** *p* < 0.01 compared with LG-untreated HaCaT cells.

**Figure 4 ncrna-09-00014-f004:**
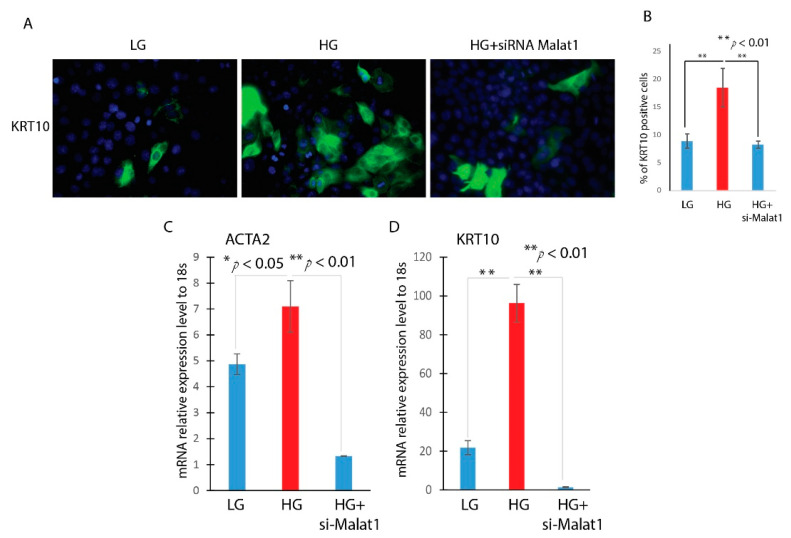
Knockdown of MALAT1 attenuates the hyperglycemia-induced EMT in HaCaT cells. HaCaT cells were transfected with MALAT1 SiRNA (Si-MALAT1) or negative control SiRNA (Si-Con) and were treated LG or HG. (**A**) The expressions of KRT10 were detected by immunofluorescence. (**B**). Quantitative analysis of number of K10-positive (K10 staining) per 20× field. (**C**,**D**). The expression level of EMT-related markers (ACTA2 and KRT10) was detected by RT-qPCR. Comparison was performed between LG, and HG, or HG treated with si-MALAT1. n = 5; mean ± SD; * *p* < 0.05, ** *p* < 0.01.

**Figure 5 ncrna-09-00014-f005:**
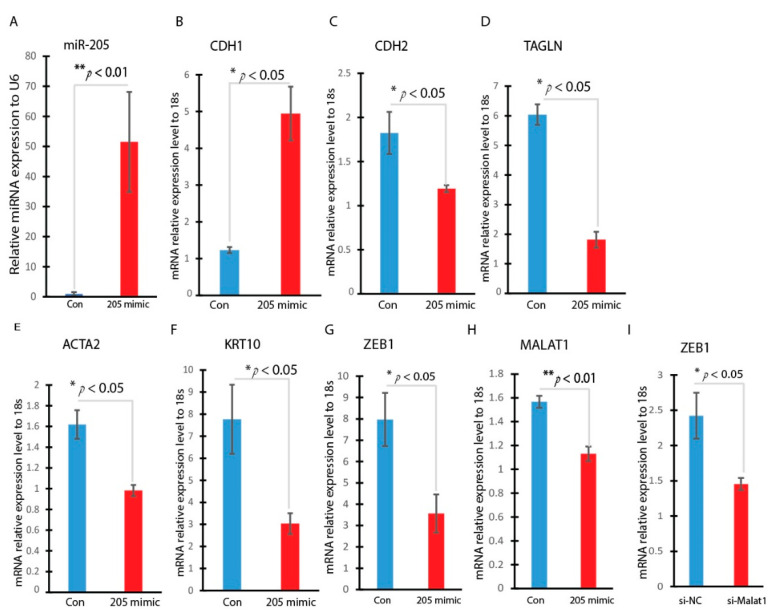
Overexpression of miR-205 inhibits the hyperglycemia-induced EMT in HaCaT cells. HaCaT cells were treated with miR-205 mimic or control mimic. RNAs were isolated and the expressions of miR-205 (**A**), CDH1 (**B**), CDH2 (**C**), TAGLN (**D**), ACTA2 (**E**), KRT10 (**F**), ZEB1 (**G**), MALAT1 (**H**) and and ZEB1 (**I**) were determined by RT-qPCR. n = 3; mean ± SD. * *p* < 0.05, ** *p* < 0.01.

**Figure 6 ncrna-09-00014-f006:**
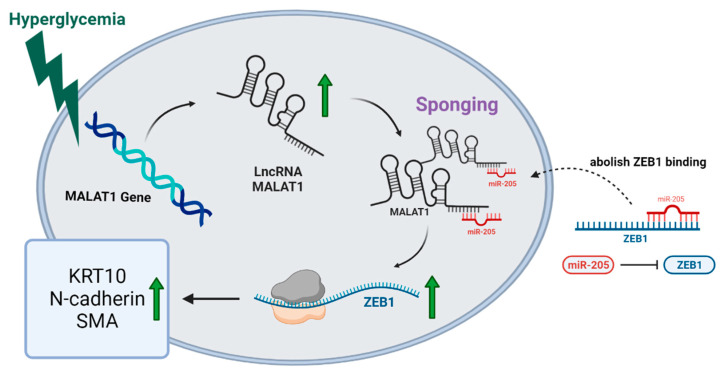
Schematic representation of conclusions drawn from this study showing that MALAT1 regulated the development of EMT in diabetic wounds through the miR-205/ZEB1 pathway.

## Data Availability

No data set generated in this study.

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
