# Peer review of "LncRNA MALAT1 Regulates Hyperglycemia Induced EMT in Keratinocyte via miR-205"

_ncrna, 2023, doi:10.3390/ncrna9010014_

Round 1

Reviewer 1 Report

The authors present a novel mechanism by which MALAT1 promotes EMT in keratinocytes, that may be associated with hyperglycemia. They propose MALAT1 acts as a competitor for miR-205 to relieve inhibition of ZEB1 and promote EMT. They use qPCR to quantify shifts in MALAT1, miR-205 and EMT related genes in WT/hyperglycemia, WT/MALAT1 knockdown, and WT/miR-205 over expression in support of their model. 

Major Comments:

The authors must present the number and location of miR-205 binding sites within MALAT1 and ZEB1 as this is a core aspect of their model.

To prove their model, the authors should show the effect of their miR-205 mimic on MALAT1 expression (expect downregulation if their model is true) and the effect of siRNA Malat1 on ZEB1 expression (expect downregualtion if their model is true).

Figure 6 is unclear, it appears to show that MALAT1 inhibits ZEB1 which is the reverse of what is stated in the paper. Rather miR-205 inhibits ZEB1 and MALAT1 acts as a competitor for miR-205 thus relieving the inhibition of ZEB1. 

The discussion makes mention of results for SMAD4 in addition to ZEB1 which are not presented in the manuscript. These results should be included.

Minor Comments:

Exact p-values (rounded to 1 significant figure) should be provided for comparisons in all figures since sufficient space is available.

Authors must specify if they are using Spearman or Pearson correlations, and provide p-values or confidence intervals for their correlations in results paragraph 1.

Typos: 

"After incubated for 24 h,"

"its morphological change to a spindle-shaped cells"

"siRNA (Si-Con) after starved for 16 h."

Capitalization of HaCat is inconsistent (HaCat vs HaCaT) please be consistent in naming this cell line.

Figure 1: Caption typo: ** P < 0.01; *** P < 0.01

Reviewer 2 Report

Please explain clearly why EMT is a detrimental mechanism in diabetic wound healing.

The authors need to show the functional effect of MALAT1 loss or miR-205 over-expression - does it impact wound healing?

Other than MALAT1, what other genes can regulate miR-205 expression? The authors must clarify the specificity of the lncRNA-miRNA interaction.

Please correct the extensive grammatical and typographical errors present in the manuscript.

Round 2

Reviewer 1 Report

While the authors have answered my questions, they have not added the information to the manuscript in a sufficiently detailed manner. To identify the binding site for miR-205 they merely state they used "bioinformatic analysis" that is not a useful description and is insufficient for a published manuscript. Where did the binding motif come from? What tool was used to find the binding site? Did the authors scan the entire gene and only find 1 binding site? Did they only scan the 3'/5'UTR and only report the first one they found?

Reviewer 2 Report

The authors have responded to the comments satisfactorily. I have no further comments

Author Response

NA